# Effects of Yeast Culture Supplementation in Wheat–Rice-Based Diet on Growth Performance, Meat Quality, and Gut Microbiota of Growing–Finishing Pigs

**DOI:** 10.3390/ani12172177

**Published:** 2022-08-25

**Authors:** Yan Lin, Chenglong Yu, Zhao Ma, Lianqiang Che, Bin Feng, Zhengfeng Fang, Shengyu Xu, Yong Zhuo, Jian Li, Junjie Zhang, Min Yang, Peng Chen, De Wu

**Affiliations:** 1Key Laboratory of Animal Disease-Resistance Nutrition and Feed Science, Institute of Animal Nutrition, Sichuan Agricultural University, Chengdu 611130, China; 2Key Laboratory of Animal Disease-Resistance Nutrition, Ministry of Education, Chengdu 611130, China; 3College of Life Science, Sichuan Agricultural University, Ya’an 625014, China; 4Pet Nutrition and Health Research Center, Chengdu Agricultural College, Chengdu 611130, China; 5Beijing Enhalor International Tech Co., Ltd., Beijing 100081, China

**Keywords:** growing–finishing pigs, meat quality, yeast culture

## Abstract

**Simple Summary:**

In recent years, the soaring price of corn–soybean meal has led to an increase in the price of feed, which has brought significant challenges to the pig industry. The addition of more grains to pig diets will help develop new feed resources and improve profits. In this study, we focused on the effects of yeast culture (YC) supplementation on growth performance, meat quality, and gut microbiota of growing–finishing pigs. Three experimental diets were compared: a corn–soybean-based diet (CON), a wheat–rice-based diet (GRA), and a wheat–rice-based diet supplemented with YC. The growth performance of pigs on the wheat–rice-based diet was similar to that of pigs on the corn–soybean-based diet, but the feed cost reduced by 6.7% compared with CON. The dressing percentages of pigs in GRA and YC also increased by 1.43% and 2.15%, respectively. Meat color and antioxidant capacity improved after YC supplementation. YC supplementation promotes the proliferation of probiotics, inhibits the growth of harmful bacteria, and improves intestinal health. Application of a cereal-based diet and YC supplementation in growing pigs appears to be a very promising strategy for cost saving and economic benefits for the swine industry.

**Abstract:**

The aim of this study was to investigate the effects of yeast culture (*Saccharomyces cerevisiae*) supplementation on the growth performance, meat quality, gut health, and microbiota community of growing–finishing pigs. A total of 45 growing–finishing pigs were randomly allocated to three treatments: a corn–soybean-based diet (CON, *n* = 15), a wheat–rice-based diet (GRA, *n* = 15), and GRA supplemented with 500 mg/kg yeast culture (YC, *n* = 15). The results show that compared to the CON group, the GRA group exhibited no significant differences in feed intake, daily gain, or feed conversion ratio, but had significantly reduced feed cost per kilogram BW gain of the finishing pigs (*p* < 0.05). Compared to that of the CON group, the GRA and YC groups showed an increase in the dressing percentage (*p* < 0.1). The meat color redness of the YC group increased (*p* < 0.1), whereas the b* value at 24 h decreased (*p* < 0.1). Meanwhile, the addition of YC significantly increased total superoxide dismutase activity on day 30 and catalase activity on day 60 (*p* < 0.05), and decreased serum urea nitrogen content on day 60 (*p* < 0.05). Furthermore, YC supplementation increased the gene expression of the duodenal anti-inflammatory factor IL-10 (*p* < 0.05), while it significantly decreased the gene expression of the ileal pro-inflammatory factor IL-8 (*p* < 0.05). The intestinal microbial identification results show that compared to the CON group, the YC group showed an increase in the relative abundances of *Lactobacillus*, *Streptococcus*, and *Clostridium* in the colon, and a decrease in the relative abundances of *Bacteroidea*, *Clostridae*, and *Prevotella* in the cecum. In conclusion, the growth performance of pigs on a wheat–rice-based diet was similar to that of pigs on a corn–soybean-based diet. Supplementation of 0.5% YC in the wheat–rice-based diet could improve the dressing percentage and meat color of growing–finishing pigs, which might be due to the increase in nitrogen utility and antioxidant capacity, and the improvement of the immune system and changes in microbiota communities.

## 1. Introduction

Yeast and yeast products, as high-quality protein sources, have been shown to improve growth performance, modulate the gut microbiome, and reduce post-weaning diarrhea in weaned piglets [1,2]. However, growing–finishing pigs have mature intestines, and the application of and research into yeast in growing–finishing pigs is less than that in piglets. Recent research has shown that yeast products have a positive effect on meat quality, as an economically important trait. Namted et al. [3] found that autolyzed yeast supplementation in pigs showed an increase in the protein percentage and a* value (redness) in the meat but reduced feed intake. Active dry yeast (ADY) and yeast cultures (YC) improve beef tenderness, while ADY improves the growth performance and carcass traits of bulls [4]. However, Zhang et al. [5] reported that although growing–finishing pigs fed a brewer’s yeast hydrolysate diet showed linear improvement in average daily body gain and feed conversion ratio, there was no effect on the carcass weight, backfat, or lean muscle percentage of pigs. Similarly, yeast culture supplementation in broiler chickens has no effect on meat quality, nutrient utilization, or bacterial count [6]. Possible reasons for these differences are related to the composition and functional properties of the different yeast products.

Wheat, rice, and corn are widely available, and wheat-based and corn-based diets are commonly used in commercial swine farms. The nutrient composition and non-starch polysaccharides of wheat and maize are relatively different. Research has shown that there were no significant differences in growth performance and pork quality in growing–finishing pigs between pigs fed a wheat-based diet and those fed a corn–soybean meal diet [7]. However, there are few studies on the effects of yeast culture supplementation on pig growth and pork quality in wheat–rice-based diets. Furthermore, supplementation with YC in growing–finishing pigs improved intestinal development and morphology and increased the relative abundance of probiotics [8]. Dietary YC supplementation in sows increased the milk lactose, protein and DM content, and changed the fecal microbiota of sows, resulting in improved growth performance of piglets [9,10]. Sun et al. [11] reported that mixed yeast culture supplementation increased body weight gain, digestibility of dry matter, and population of *Lactobacillus* in broilers. These results suggest that yeast and yeast products have important effects on intestinal health and microbial flora. Therefore, this study was conducted to test the hypothesis that YC can improve pork quality by improving intestinal health and microbial composition. Thus, the aim of this study was to determine the effects of corn and wheat–rice-based diet, and YC (*Saccharomyces*
*cerevisiae* (*S. cerevisiae*), CGMCC No. 6120) supplementation on growth performance, antioxidant capacity, intestinal development and flora in growing–finishing pigs fed wheat–rice-based diets.

## 2. Materials and Methods

The experimental protocol was approved by the Sichuan Agricultural University Animal Care and Use Committee (SAU-ANI-2021-106). 

### 2.1. Animals and Experimental Design

A total of 45 crossbred ((Yorkshire × Landrace) × Yorkshire) pigs (50.75 ± 1.319 kg body weight (BW)) were randomly allotted to one of three groups. The temperature of room was maintained around 23–27 °C. The pen was a slatted concrete floor facility equipped with a feeder and a water nipple to provide feed and water *ad libitum*. Pigs were fed daily at 800, 1400, and 2000 h. Pigs had *ad libitum* access to food and water during 60 days trial. BW was measured every 30 days, and feed intake was recorded daily. 

### 2.2. Diets and Yeast Culture 

*S**accharomyces cerevisiae* yeast culture with moisture 4.87%, CP 18.41%, ash 7.69%, β-glucan and mannan oligosaccharides 0.82%, and others 69.03% (Enhalor Biotechnology Company, Beijing, China). The diets were control (CON; corn–soybean meal diets), wheat–rice (unhusked)-based diets (GRA), and GRA + 0.5% YC-supplemented diet (YC). Experimental diets were formulated to meet or exceed the predicted requirements for growing–finishing pigs. All pigs were fed a mash diet according to the three-phase feeding program (Table 1). 

### 2.3. Sample Collection and Chemical Analyses

Fasting blood samples were collected from the anterior vena cava on days 30 and 60 in disposable culture tubes and centrifuged for 10 min at 3000 rpm at 4 °C. The supernatant was stored at −20 °C. Serum glucose (Glu), albumin (ALB), total protein (TP), and blood urea nitrogen (BUN) levels were detected using an automatic biochemical analyzer (Hitachi 7600-110R, Hitachi Co., Hitachi, Yokohama, Japan). Total antioxidant capacity (T-AOC), total superoxide dismutase (T-SOD) and catalase (CAT) activities, and malondialdehyde (MDA) level were measured using commercial kits (Nanjing Jiancheng Bioengineering Institute, Nanjing, China). 

Six female pigs per group were selected and slaughtered by electronarcosis and exsanguination, based on standard commercial procedures. Carcass weight and tare weight were measured and used to calculate the dressing percentage. Average backfat thickness was calculated as the average of the first rib, the last rib, and the last lumbar vertebra. Eye muscle area (EMA) was measured using Vernier calipers [12]. Carcass length was measured using dressmaker tape with an accuracy of 1 mm. The heart, liver, spleen, kidney, lung, and abdominal fat were immediately removed, properly cleaned, and individually weighed (g). The organ index = (organ weight in grams)/(live weight, in grams) × 100.

The longissimus dorsi muscle from the 10th rib was used to determine meat color, pH, marbling score, dripping loss, and cooking loss, according to the method described by Chen [12]. Muscle pH_45min_ and pH_24h_ were calculated using a pH meter (pH-STAR, SFK-Technology, Copenhagen, Denmark), which was calibrated with pH 4.6 and pH 7.0 buffers. Meat color (L* lightness, a* redness, and b* yellowness) was recorded 45 min and 24 h after slaughter using a portable chromameter (CR-410, Kinica Minolta Sensing Inc., Minolta, Japan). The marbling score at 24 h was measured using the National Pork Producer Council standards (NPPC; Des Moines, IA, USA). 

### 2.4. Intestinal Morphology

Intestines fixed in 4% paraformaldehyde were prepared based on the conventional paraffin-embedding techniques. In brief, cross-sections of the segments were cut into 5-μm-thick slices, then stained with hematoxylin and eosin (H&E) (Servicebio Technology Co., Ltd., Wuhan, China). In each sample, 15 intact, well-oriented crypt villus units were chosen. Villus height and crypt depth (CD) were measured using Image-Pro Plus 6.0 (Media Cybernetics Inc., Bethesda, MD, USA).

### 2.5. Bacterial Community Analysis

To determine the microbial composition, 16S rDNA sequencing was used. Thirty-six fresh content samples of colon and cecum collected from six pigs in each group were used to evaluate the microflora communities. The total DNA was extracted from the samples using the E.Z.N.A. soil DNA kit (Omega Bio-tek, Norcross, GA, USA) according to the manufacturer’s instructions, and the concentration of the total DNA was determined using a NanoDrop 2000 UV–Vis spectrophotometer (Thermo Scientific, Wilmington, NC, USA). Extracted DNA samples were sent to Majorbio Bio-Pharm Technology Co. Ltd. (Shanghai, China) for amplicon pyrosequencing using the Illumina MiSeq PE300 platform (Illumina, San Diego, CA, USA). The V3–V4 hypervariable region of the bacterial 16S rRNA gene was amplified using the 338F and 806R primers (5’-ACTCCTACGGGAGGCAGCAG-3’ and 5’-GGACTACHVGGGTWTCTAAT-3’, respectively) using an ABI GeneAmp 9700 PCR thermocycler (Applied Biosystems, Foster City, CA, USA). Operational taxonomic units (OTUs) with a 97% similarity cutoff were clustered using UPARSE version 7.1 [13]. The taxonomy of the representative sequence of each OTU was analyzed using RDP Classifier version 2.2 [14] against the 16S rRNA database (SILVA v138) with a 70% confidence threshold. 

### 2.6. Real-Time Quantitative PCR

Total RNA was isolated and purified using a total RNA extraction kit (TaKaRa, Dalian, China). Reverse transcription to synthesize cDNA was performed using the PrimeScript RT reagent kit with gDNA Eraser (TaKaRa, China). Table 2 contains the probes and primers used for each gene. Fluorescence-based quantitative polymerase chain reaction (PCR) assays were conducted using a 7900HT Real-Time PCR system (Applied Biosystem, CA, USA) with the following cycle profile: 94 °C for 30 s, followed by 40 cycles of 94 °C for 5 s, and 60 °C for 30 s. Each sample was analyzed in triplicate. Relative expression was expressed as the ratio of the target gene to β-actin mRNA according to the 2^−∆∆Ct^ method [15].

### 2.7. Statistical Analysis 

Data were analyzed using the one-way ANOVA procedure of the Statistical Product and Service Solutions (SAS) statistical software (V9.4; SAS Institute Inc., Cary, NC, USA), followed by a generalized linear model (GLM) analysis and Tukey’s tests. Values are expressed as mean ± SEM, with differences being considered statistically significant at *p* < 0.05 and a value of 0.05 ≤ *p* < 0.10 being indicative of a ‘tendency’. 

## 3. Results

### 3.1. Growth Performance and Cost 

The effects of the diet types and YC supplementation on growth performance are shown in Table 3. No differences were observed in BW, average daily gain (ADG), or average daily feed intake (ADFI) among the groups. When the price of wheat was 85–88% of that of corn, the weight gain cost per kilogram of GRA was significantly lower than that of CON during 0–31 days and 0–61 days (*p* < 0.05).

### 3.2. Carcass Characteristics and Organ Development

Compared to that of the CON group, the YC group tended to exhibit an increase in the dressing percentage by 2.9% (Table 4, *p* < 0.1), while there was no significant difference between the GRA and the YC groups (*p* > 0.05). YC supplementation of the GRA decreased the kidney index (*p* < 0.05) and had no effect on the development of other organs (*p* > 0.05).

### 3.3. Meat Quality

As shown in Table 5, diet type and YC supplementation had no effect on meat quality, except for the Hunter values. Compared to CON, YC improved the meat quality of finishing pigs, increased the muscle a* value, and decreased the muscle b* value at 24 h (*p* < 0.10). 

### 3.4. Blood Metabolites 

As shown in Table 6, diet type and YC supplementation had no significant effects on the serum parameters on day 30. However, compared to CON, dietary YC supplementation significantly decreased TP, ALB, and urea nitrogen levels on day 60 (*p* < 0.05). 

### 3.5. Serum Antioxidant Capacity

As shown in Table 7, compared to the corn–soybean meal diet, the wheat–rice-based diet increased the T-SOD activity on day 30 in serum, while dietary YC supplementation not only increased CAT activity in serum, but also decreased the MDA level in serum (*p* < 0.05). 

### 3.6. Intestinal Morphology 

Intestinal morphology results are presented in Figure 1 and Table 8. Compared to that of the CON group, the ratio of the villous height to crypt depth of jejunum in the GRA and YC groups increased by 15% and 23%, respectively, although there was no significant difference (*p* > 0.05). However, the VH and CD were not significantly altered in the ileum (*p* > 0.05). 

### 3.7. Gene Expression 

As shown in Figure 2, compared to that of the CON group, the expression of IL-1β in the duodenum was downregulated in the YC group (*p* < 0.01). Moreover, the expression of IL-8 and IL-10 was significantly upregulated in the GRA and YC groups (*p* < 0.05), and the relative expression of IL-10 mRNA in the YC group tended to increase (*p* < 0.05). However, there was no effect on the expression of IL-6, TNF-α, occludin, claudin, or zonula occludens-1 (ZO-1) among the groups (*p* > 0.05).

As shown in Figure 3, compared to that of the CON group, the expression of IL-10 in the jejunum was upregulated in the GRA and YC groups (*p* < 0.05). Furthermore, the expression of IL-10 in the YC group was higher than that in the GRA group, whereas the expression of IL-1β was significantly downregulated in the YC group (*p* < 0.05).

As shown in Figure 4, compared to that of the CON group, the expression of IL-1β in the ileum was downregulated in the GRA group (*p* < 0.05), and the expression of IL-8 was also significantly downregulated in the GRA and YC groups (*p* < 0.05). The relative mRNA expression of claudin and IL-10 in the YC group was significantly upregulated (*p* < 0.05). 

### 3.8. Overview of 16S rRNA Sequencing Data 

To explore the effect of dietary yeast culture on gut microbiota, chyme samples were collected for 16S rRNA sequencing. In total, 1,423,816 sequences and 594,334,820 bases were obtained. The average length of the sequence was 417 bp.

Furthermore, the microbial composition of the colonic chyme was analyzed to observe alterations in the bacterial community composition. In brief, at the phylum level (Figure 5A), the dominant phyla were Firmicutes and Bacteroidetes, and the abundance of Firmicutes in the GRA (83.98%) and YC (84.33%) groups was higher than that in the CON group (72.11%). At the order level (Figure 5B), the abundance of Lactobacillales in the GRA (50.51%) and YC groups (41.88%) was higher than that in the CON group (24.44%) (*p* < 0.05), whereas the abundance of Bacteroidetes (22.20%) in the CON group was higher than that in the GRA (11.51%) and YC (11.60%) (*p* < 0.01) groups. At the family level (Figure 5C), the abundances of *Prevotellaceae*, *Selenomonadaceae*, *Christensenellaceae*, *Butyricicoccaceae*, and *UCG-010* in the GRA and YC groups were higher than those in the CON group (*p* < 0.05). At the genus level (Figure 5D), the abundance of *Prevotella* was higher in the CON and GRA groups than in the YC group (*p* < 0.05). Moreover, the abundances of *Prevotellaceae_NK3B31_group*, *Christensenellaceae_R-7_group*, *Anaerovibrio*, and *norank_f__UCG-010* were higher in the CON group than in the GRA and YC groups. 

Dietary YC supplementation changed the microbial composition of cecal content. Figure 6A showed that Firmicutes and Bacteroidetes were the dominant phyla in all groups, and the abundance of Firmicutes in the GRA (76.99%) and YC (80.75%) groups was higher than in the CON group (71.11%). At the order level (Figure 6B), the abundance of RF39 in the CON and YC groups was higher than in the GRA group (*p* < 0.05), whereas the abundance of Peptococcales in the GRA group was higher than in the CON group (*p* < 0.05). At the family level (Figure 6C), the abundances of *Selenomonadacea* and *Butyricicoccaceae* in the CON group were higher than in the YC group (*p* < 0.05). The abundance of *norank_o__Coriobacteriales* in the YC group was higher than in the CON group (*p* < 0.05), whereas the abundances of *norank_o__RF39* in the CON and YC groups were higher than in the GRA group (p < 0.05). At the genus level (Figure 6D), the abundances of *norank_f__norank_o__Coriobacteriales* in the YC group were higher than in the CON group (*p* < 0.05), whereas the abundances of *Blautia* and *Peptococcus* in the GRA group were higher than in the CON group (*p* < 0.05), and the abundances of *Prevotellaceae_NK3B31_group*, *Turicibacter*, and *Lachnospira* in the CON group were higher than in the GRA group (*p* < 0.05).

## 4. Discussion

These findings are useful for the rational utilization of feed resources in the swine industry. There are significant differences in nutritional composition between wheat–rice-based and corn-based diets. In our study, we found that the type of diet and YC supplementation had no significant effect on the growth performance of growing pigs in the isoenergetic and isonitrogen models. A previous study reported that diet type did not influence ADG, but growing pigs fed corn–soybean meal diets from days 0 to 64 had decreased ADFI and increased G:F compared to the pigs fed 30% dried distillers grains with solubles and 19% wheat middling diets during this period [16]. The apparent ileal digestibilities of dry matter, neutral detergent fiber, and gross energy (GE) of growing pigs fed a corn-based diet were similar to those of pigs fed a wheat-based diet [17]. Weaned pigs that were fed maize-based diets had a 60 g higher ADG than pigs that were fed wheat–barley-based diets, even though the daily feed intake was similar because of higher N retention [18]. Corn-based and wheat-based diets had no effect on ADG, ADFI, or final BW over the entire 4-week study period or final BW in weaned piglets [19]. Navarro et al. [20] found that the apparent total tract digestibility of GE in growing pigs was higher in pigs fed wheat-based diets than in pigs on DDGS diets, but was similar to that of the pigs fed corn-based diets. Dietary administration of 3 g/kg yeast hydrolysate improved the growth performance of growing–finishing pigs [21]. Although supplementation with YC had no effect on growth performance, it resulted in better intestinal development and morphology. Mayorga et al. [22] reported that dietary live yeast did not affect growth performance; however, it tended to improve G:F. Furthermore, when the price of wheat and rice was 13% lower than that of corn, the weight gain cost per kilogram was reduced by 6–7%.

Different carbohydrate sources in the diet may have different effects on meat quality because of the differences in starch composition and structure. Interestingly, the dressing percentage of pigs fed GRA and YC increased by 1.93% and 2.90%, respectively, compared to that of the CON group. This may be related to an improvement in nitrogen utilization. In this study, we found that dietary YC supplementation significantly decreased the TP, ALB, and urea nitrogen levels on day 60. Meanwhile, Namted et al. [3] reported that BUN decreased with the addition of 0.5% autolyzed yeast. Pigs fed diets containing mycotoxins and yeast cell wall extract showed increased apparent ileal digestibility of crude protein, and the GE of nursery piglets [23]. Dietary live yeast supplementation during gestation and lactation, and during the nursery phase, increased nutrient digestibility in the offspring [24]. Similarly, yeast hydrolysate supplementation in broiler chickens increased the eviscerated yield rate and chest muscle yield [25]. In addition to the role of yeast, diet type may also affect the dressing percentage. Although there are few studies on the effects of dietary types on pork carcass quality, Li et al. [26] reported that compared to a waxy maize starch diet, a pea starch diet increased the loin eye area and pH_45min_ value of finishing pigs. Ran et al. [27] reported that lambs fed a sorghum-based diet had lower dressing percentage and meat quality than those fed a corn-based diet. This implies that the different effects of carbohydrate sources on pork quality may be related to starch structure and composition. More in-depth and detailed research is required in the future. 

Furthermore, the GRA-based diet and YC tended to improve the meat quality of finishing pigs. Namted et al. [3] reported that 0.5% autolyzed yeast in the diet of pigs increased their springiness and tended to increase the redness and pH of the meat at 6 h postmortem. The improvement in beef tenderness following active dry yeast supplementation of finishing bulls is related to an increase in the level of CAT in the serum and glutathione reductase in the meat [4]. Furthermore, dietary live yeast supplementation in broilers alleviates transport-stress-impaired meat quality [8]. Antioxidant capacity and inflammatory responses have an important impact on meat quality. In this study, the T-SOD activities in the GRA and YC groups were higher than that in the CON group, while dietary YC supplementation increased antioxidant capacity in pigs. Yeast hydrolysate supplementation in growing–finishing pigs tended to increase T-AOC activities and enhanced serum IL-10 levels [21]. Yeast hydrolysate increased SOD and GSH-Px activities in the serum and meat quality of broiler chickens [25], and enhanced the antioxidant status of juvenile Nile tilapia [28]. 

The intestine plays an important role in immunity function. In this study, YC supplementation changed the expression levels of tight-junction-related genes, improving intestinal barrier and immunity. Fu et al. [21] showed that supplementing weaned piglets with yeast hydrolysate upregulated the mRNA expression of ZO-1. The supplementation of yeast culture increased the mRNA levels of ZO-1, claudin, and occludin in juvenile largemouth bass [29]. Furthermore, a wheat–rice-based diet upregulated IL-8 and IL-10 gene expression in the intestine, while YC supplementation decreased the expression of the pro-inflammatory cytokine IL-1β and upregulated the expression of the anti-inflammatory cytokines IL-8 and IL-10 in the duodenum, jejunum, and claudin of the ileum, respectively, which implies that dietary type and YC have positive effects on the intestinal development and immunity. In fact, research has shown that dietary superfine yeast powder supplementation increased lysozyme levels at day 7 and jejunum mucosal slgA secretion, while dietary live yeast supplementation increased jejunum mucosal slgA secretion in piglets [30]. Broiler chickens fed a live yeast culture diet show decreased serum TNF-α, IL-1β, and IL-6 levels [31]. Oral treatment with yeast β-glucan in non-obese diabetic mice has positive changes to immune functions by increasing IL-10, IL-17, and IL-21 levels and decreasing TNF-α level, although levels of some pro-inflammatory cytokines (IL-1b and IFN-γ) were increased [32]. Meanwhile, β-glucan from *S. cerevisiae* ameliorated inflammation in a DSS-induced mouse model of colitis by regulating the levels of cytokines (IL-1β, IL-6, TNF-α, and IFN-γ), increasing the expression levels of tight junction proteins [33], implying that yeast culture has favorable immunomodulatory activity. 

Gut microbial communities play a considerable role in metabolism and immunity [34]. However, there is minimal research on gut microbes and meat quality. Research has shown that an 80% feed restriction in bearded chickens could improve the meat quality and flavor by changing the lipid metabolism and the structure of the cecal microbial community [35]. Black pepper extract supplementation in finishing pigs linearly increased fecal *Lactobacillus* counts and backfat thickness at week 10 [36]. Zhu et al. [37] found that there is a significant positive correlation between the abundance of the ileal genera *Streptococcus* and L* value at 24 and 48 h, and a significant negative correlation between the abundance of unidentified_*Ruminococcasceae* in both the ileum and colon and L* value at 24 h. Furthermore, the gut microbiota of obese Jinhua pigs intrinsically promotes intramuscular fat accumulation and can be transferred to mouse recipients by transplanting fecal microbiota [38]. The aforementioned research shows that the microbial composition has a significant impact on pork quality. In this study, we found that yeast culture supplementation tended to increase the relative abundance of probiotics, such as *Lactobacillus, Streptococcus*, and *Clostridium,* decreasing the relative abundance of pathogenic microorganisms, such as *Bacteroides* and *Prevotella* in the cecum, thereby increasing the meat quality. In fact, in vitro research has shown that a live yeast probiotic (*S. cerevisiae*) can improve microflora composition in a colonic simulation model of pigs [39]. *Saccharomyces cerevisiae* yeast-supplemented horses showed increased abundances of *Lactobacilli* and lactic acid utilizers in the cecum but remained similar in the colon, and the abundance of *Lactobacilli* for the high-starch diet was higher than that in the for high-fiber diet in the cecum and colon [40]. Furthermore, hydrolyzed yeast supplementation in weaned piglets increased the *Lactobacillus* spp. count and acetate and propionate production [1]. The supplementation with 1.0% brewer’s yeast hydrolysate can improve ADFI and G:F, which may be related to the fecal *Escherichia* microbial counts and diarrhea in weanling pigs [41]. Supplementation with the live yeast *S. cerevisiae* significantly affected microbial diversity in cecal contents and short-chain fatty-acid-producing bacteria in suckling piglets [42]. The supplementation with yeast culture increased the abundance of probiotics (such as *Lactobacillus*, *Bacillus*, and *Bifidobacterium*) and decreased the abundance of intestinal potential pathogenic bacteria (*Plesiomonas*) in juvenile largemouth bass [29]. Supplementation with live yeast cultures increased the abundance of the phyla Firmicutes and genera *Lactobacillus*, *Prevotella*, and *Enterococcus* in broiler chickens [31]. In addition, studies on humans and pigs have shown that diet type significantly affects the intestinal microflora. Lower vegetable intake mediates differences in the abundances of *Lachnospira* and *Ruminococcus-1*, and higher red meat intake mediates differences in the abundances of *Lachnospira* and *Ruminococcus-1* [43]. Pea starch supplementation in growing barrows increased the relative abundance of amylolytic bacteria, such as *Lactobacillus* spp. and *Streptococcus* spp., and decreased the relative abundance of some inflammatory bacteria, such as *Tyzzerella*, *Porphyromonas*, and *Tannerella* in the cecum of pigs. Thus, a high ratio of amylose to amylopectin in diets may be beneficial for the intestinal health of pigs [44].

## 5. Conclusions

Compared with the corn–soybean meal diet, the wheat–rice diet had a similar effect on the growth performance of growing pigs. If the price of wheat–rice is 90% lower than of the price of corn, the application of wheat–rice-type diet will greatly save the feed cost. Meanwhile, yeast culture supplementation in wheat–rice-type diet positively increased the dressing percentage and meat color by increasing nitrogen utility, antioxidant capacity, and modulating the immune system and microbiota communities of growing–finishing pigs. 

## Figures and Tables

**Figure 1 animals-12-02177-f001:**
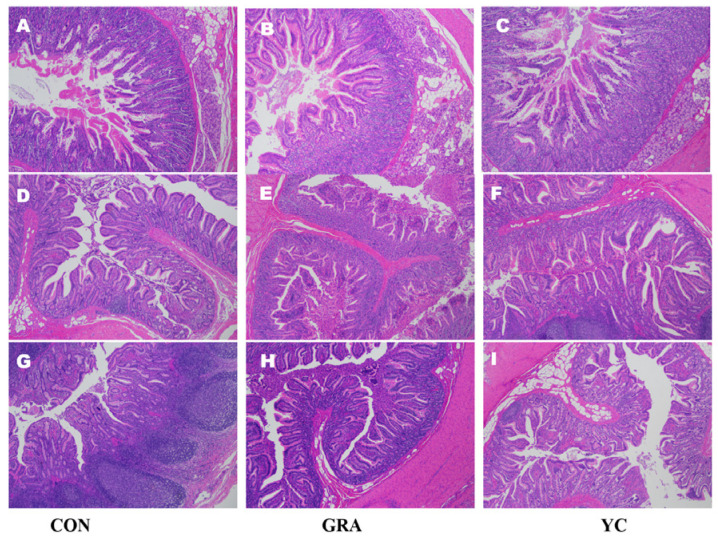
Effect of yeast culture supplementation on intestinal histology of finishing pigs. (**A**–**C**) Duodenum; (**D**–**F**) jejunum; (**G**–**I**) ileum. Figure presented at 200-fold field of vision, *n* = 6 per group.

**Figure 2 animals-12-02177-f002:**
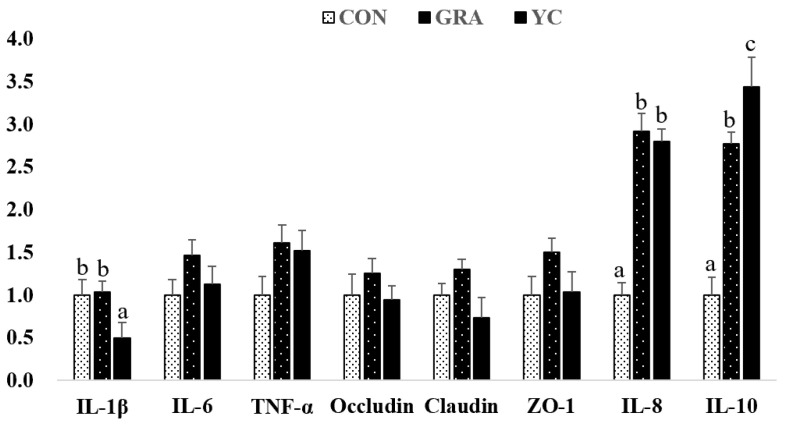
Effect of yeast culture supplementation on the mRNA levels of immunity-related genes in duodena of finishing pigs. IL-10, interleukin 10; TNF-α, tumor necrosis factor-α; IL-1β, interleukin-1β; IL-6, interleukin-6; IL-8, interleukin-8; ZO-1, zonula occludens-1. Values are means and SEMs, *n* = 6 per group. a,b,c *p* < 0.05 between different superscripts within the same gene.

**Figure 3 animals-12-02177-f003:**
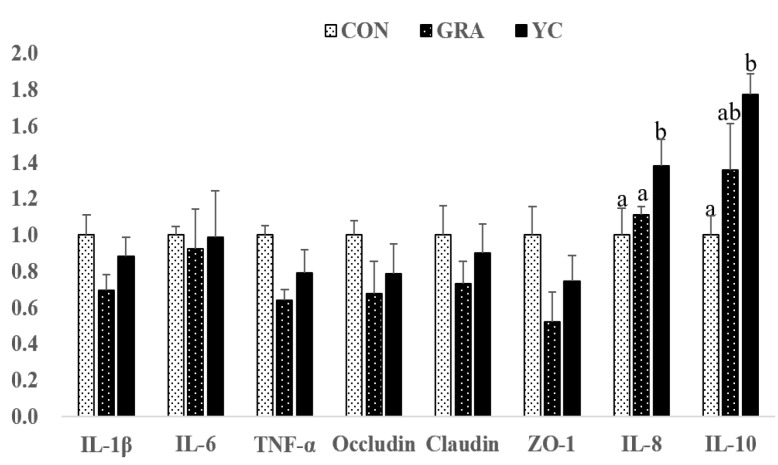
Effect of yeast culture supplementation on the mRNA levels of immunity-related genes in jejuna of finishing pigs. IL-10, interleukin 10; TNF-α, tumor necrosis factor-α; IL-1β, interleukin-1β; IL-6, interleukin-6; IL-8, interleukin-8; ZO-1, zonula occludens-1. Values are means and SEMs, *n* = 6 per group. a,b *p* < 0.05 between different superscripts within the same gene.

**Figure 4 animals-12-02177-f004:**
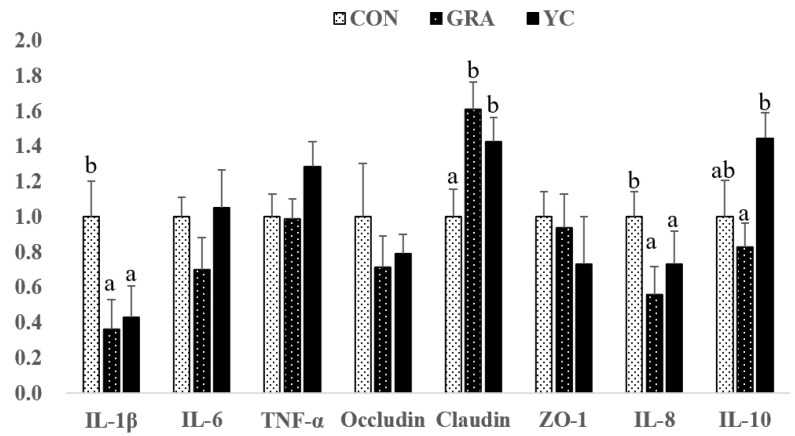
Effect of yeast culture supplementation on the mRNA levels of immunity-related genes in ilea of finishing pigs. IL-10, interleukin 10; TNF-α, tumor necrosis factor-α; IL-1β, interleukin-1β; IL-6, interleukin-6; IL-8, interleukin-8; ZO-1, zonula occludens-1. Values are means and SEMs, *n* = 6 per group. a,b *p* < 0.05 between different superscripts within the same gene.

**Figure 5 animals-12-02177-f005:**
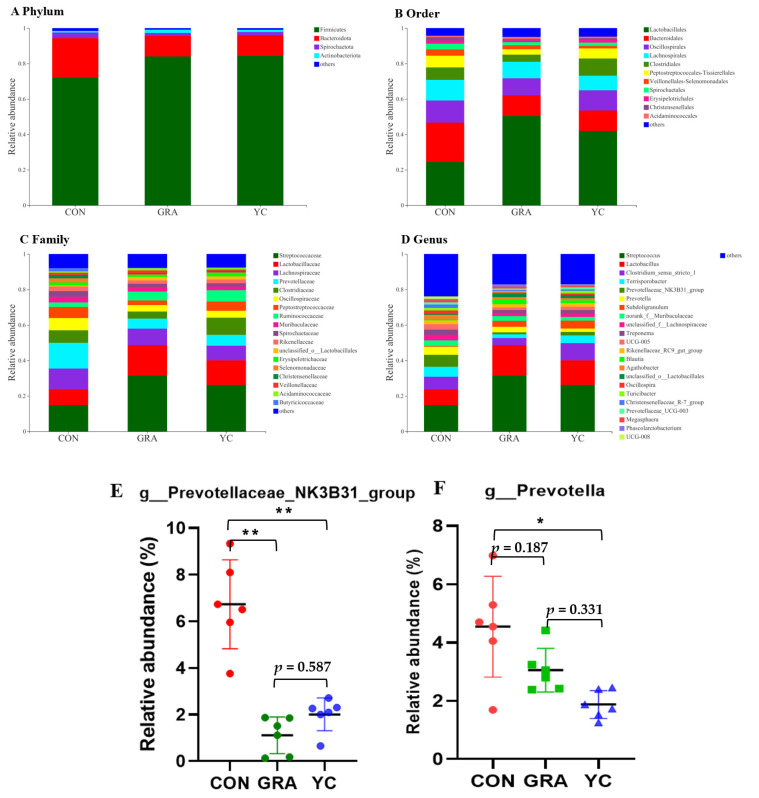
Effects of yeast culture on colon chyme microbial composition. (**A**) The relative amounts of chyme microbiota at the phylum level. (**B**) The relative amounts of chyme microbiota at the order level. (**C**) The relative amounts of chyme microbiota at the family level. (**D**–**F**) The relative amounts of chyme microbiota at the genus level. Data are expressed as the mean ± SEM. * *p* < 0.05 and ** *p* < 0.01.

**Figure 6 animals-12-02177-f006:**
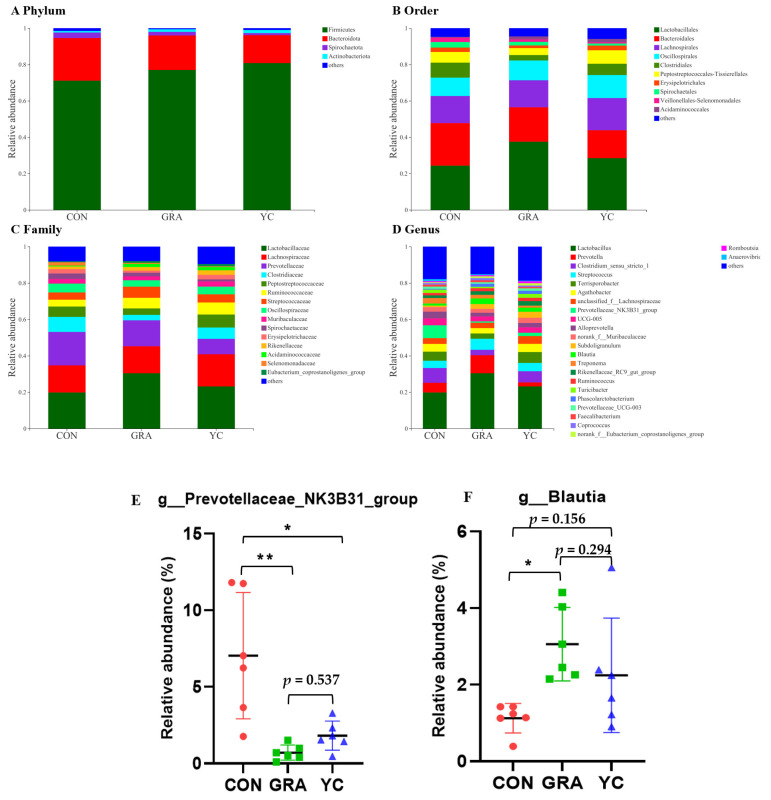
Effects of yeast culture on cecum chyme microbial composition. (**A**) The relative amounts of chyme microbiota at the phylum level. (**B**) The relative amounts of chyme microbiota at the order level. (**C**) The relative amounts of chyme microbiota at the family level. (**D**–**F**) The relative amounts of chyme microbiota at the genus level. Data are expressed as the mean ± SEM. * *p* < 0.05 and ** *p* < 0.01.

**Table 1 animals-12-02177-t001:** Ingredient and nutrient values of the experimental diet (%).

Ingredients, %	50–75 kg			75–100 kg			100–110 kg		
	CON	GRA	YC	CON	GRA	YC	CON	GRA	YC
Corn	74.59	10.00	10.00	74.90	0.00	0.00	79.90	0.00	0.00
Wheat	0.00	55.87	55.37	0.00	55.00	54.50	0.00	59.20	58.70
Unhusked rice	0.00	16.40	16.45	0.00	27.50	27.50	0.00	29.03	29.07
Soybean oil	1.01	2.00	2.00	1.00	2.00	2.00	1.00	2.00	2.00
Soybean meal	21.24	12.60	12.55	21.40	12.87	12.87	16.62	7.41	7.37
Yeast culture	0.00	0.00	0.50	0.00	0.00	0.50	0.00	0.00	0.50
L-Lysine HCl	0.30	0.48	0.48	0.12	0.30	0.30	0.10	0.30	0.30
DL-methionine	0.03	0.05	0.05	0.03	0.05	0.05	0.00	0.00	0.00
L-Threonine	0.07	0.17	0.17	0.02	0.10	0.10	0.00	0.10	0.10
L-Tryptophan	0.01	0.00	0.00	0.00	0.00	0.00	0.00	0.00	0.00
Choline 50%	0.10	0.10	0.10	0.10	0.10	0.10	0.10	0.10	0.10
Limestone	0.64	0.80	0.80	0.60	0.80	0.80	0.50	0.78	0.78
Dicalcium phosphate	1.18	0.70	0.70	1.00	0.45	0.45	0.95	0.25	0.25
Salt	0.30	0.30	0.30	0.30	0.30	0.30	0.30	0.30	0.30
Vitamin premix1	0.03	0.03	0.03	0.03	0.03	0.03	0.03	0.03	0.03
Mineral premix1	0.50	0.50	0.50	0.50	0.50	0.50	0.50	0.50	0.50
Calculated Values									
DE, Mcal/kg	3.37	3.29	3.29	3.38	3.23	3.22	3.39	3.23	3.22
CP, %	15.75	15.77	15.77	15.63	15.62	15.63	13.87	13.86	13.86
Total Lysine	0.98	0.98	0.98	0.85	0.85	0.85	0.72	0.73	0.73
Total methionine	0.28	0.29	0.29	0.28	0.29	0.29	0.23	0.22	0.22
Total Threonine	0.65	0.67	0.67	0.60	0.60	0.60	0.52	0.52	0.53
Total Tryptophan	0.18	0.18	0.18	0.17	0.18	0.18	0.14	0.16	0.16
Ca, %	0.59	0.59	0.59	0.53	0.54	0.54	0.47	0.47	0.47
TP, %	0.55	0.52	0.52	0.52	0.49	0.49	0.49	0.44	0.44

Notes: CON, corn–soybean-based diet; GRA, wheat–rice-based diet; YC, GRA supplemented with yeast culture.

**Table 2 animals-12-02177-t002:** Primer sequences used in this study.

Gene	Primers	Accession No	Product Length (bp)
*IL-10*	F: CACTGCTCTATTGCCTGATCTTCC R: AAACTCTTCACTGGGCCGAAG	NM_214041.1	136
*TNF-α*	F: TGGCCCCTTGAGCATCA R: CGGGCTTATCTGAGGTTTGAGA	NM_214022.1	68
*IL-1β*	F: TACCCTCTCCAGCCAGTCTTCA R: AGGTCCAGGTTTTGGGTGCAG	NM_214055.1	167
*IL-6*	F: AGGGAAATGTCGAGGCTGTGC R: CCGGCATTTGTGGTGGGGTT	NM_214399.1	112
*IL-8*	F: AGGACCAGAGCCAGGAAGAGAC R: CACAGAGAGCTGCAGAAAGCAG	NM_213867.1	108
*ZO-1*	F: CAGCCCCCGTACATGGAGA R: GCGCAGACGGTGTTCATAGTT	XM_021098896.1	114
*Occludin*	F: ATGCCTCCTCCCCTTTCGGA R: CGCCCGTCGTGTAGTCTGTC	NM_001163647.2	295
*Claudin-1*	F: GATCGGCTCCA TCGTCAGCA R: CATTGACTGGGGTCATGGGGTC	NM_001244539.1	416
*β-actin*	F: TCTGGCACCACACCTTCT R: TGATCTGGGTCATCTTCTCAC	XM_021086047.1	114

**Table 3 animals-12-02177-t003:** Effect of dietary yeast culture supplementation on growth performance of finishing pigs.

Item	CON	GRA	YC	*p* Value
BW (kg)				
0 d	45.61 ± 0.98	45.76 ± 1.02	46.82 ± 0.91	0.642
30 d	79.04 ± 1.06	79.01 ± 1.31	80.21 ± 1.56	0.771
60 d	111.55 ± 0.70	111.40 ± 2.42	112.46 ± 2.36	0.921
ADG (g)				
0–30 d, g	1078.28 ± 20.56	1072.69 ± 14.73	1076.99 ± 22.69	0.978
31–60 d, g	1083.78 ± 31.27	1079.56 ± 38.19	1075.11 ± 38.19	0.987
0–60 d, g	1080.98 ± 18.56	1076.07 ± 24.19	1076.07 ± 26.13	0.985
ADFI (g)				
0–30 d	2960.69 ± 47.00	2860.75 ± 84.15	2926.50 ± 77.37	0.615
31–60 d	3340.67 ± 137.53	3389.89 ± 173.17	3269.53 ± 155.70	0.862
0–60 d	3147.57 ± 89.82	3120.98 ± 117.20	3095.20 ± 113.70	0.943
F:G ratio				
0–30 d	2.75 ± 0.05	2.73 ± 0.02	2.72 ± 0.03	0.851
31–60 d	3.09 ± 0.12	3.14 ± 0.10	3.04 ± 0.10	0.828
0–60 d	2.91 ± 0.06	2.90 ± 0.08	2.88 ± 0.05	0.928
Cost/kg of gain (US$)				
0–30 d	1.52 ± 0.03 ^a^	1.42 ± 0.01 ^b^	1.45 ± 0.02 ^ab^	0.040
31–60 d	1.78 ± 0.07	1.66 ± 0.05	1.55 ± 0.05	0.130
0–60 d	1.65 ± 0.07 ^A^	1.54 ± 0.06 ^B^	1.58 ± 0.06 ^AB^	0.080

Notes: ADG, average daily gain; ADFI, average daily feed intake; F:G, feed-to-gain ratio. Results are presented as the mean ± SEM. Small letters within the same line indicate *p* < 0.05 and capital letters indicate 0.05 ≤ *p* < 0.10.

**Table 4 animals-12-02177-t004:** Effect of yeast culture supplementation on carcass characteristics and organ development of finishing pigs.

Item	CON	GRA	YC	*p* Value
Eye muscle area, cm^2^	53.63 ± 2.66	54.39 ± 2.61	49.32 ± 1.73	0.301
Backfat thickness, mm	17.03 ± 0.47	18.96 ± 1.89	19.07 ± 1.38	0.627
Dressing percentage, %	74.15 ± 0.23 ^A^	75.58 ± 0.52 ^AB^	76.30 ± 0.31 ^B^	0.060
Carcass length/cm	83.70 ± 1.93	84.36 ± 0.86	85.68 ± 0.63	0.612
Abdominal fat index, %	1.05 ± 0.07	1.06 ± 0.09	1.15 ± 0.15	0.789
Heart index, %	0.36 ± 0.01	0.34 ± 0.01	0.34 ± 0.01	0.474
Liver index, %	1.46 ± 0.04	1.45 ± 0.05	1.43 ± 0.03	0.878
Spleen index, %	0.14 ± 0.01	0.13 ± 0.00	0.15 ± 0.01	0.370
Lung index, %	0.65 ± 0.07	0.62 ± 0.03	0.62 ± 0.07	0.912
Kidney index, %	0.31 ± 0.01 ^a^	0.28 ± 0.01 ^ab^	0.27 ± 0.01 ^b^	0.011

Notes: Results are presented as the mean ± SEM (*n* = 6). Small letters within the same line indicate *p* < 0.05 and capital letters indicate 0.05 ≤ *p* < 0.10. Abdominal fat index (%) = Abdominal fat weight/live weight × 100. Heart index (%) = Heart weight/live weight × 100. Liver index (%) = Liver weight/live weight × 100. Spleen index (%) = Spleen weight/live weight × 100. Lung index (%) = Lung weight/live weight × 100. Kidney index (%) = Kidney weight/live weight × 100.

**Table 5 animals-12-02177-t005:** Effect of yeast culture supplementation on meat quality of finishing pigs.

Items	CON	GRA	YC	*p* Value
pH_45min_	6.34 ± 0.09	6.08 ± 0.16	6.16 ± 0.18	0.475
pH_24h_	5.43 ± 0.03	5.42 ± 0.03	5.41 ± 0.02	0.888
pH_48h_	5.45 ± 0.02	5.42 ± 0.03	5.44 ± 0.02	0.727
Drip loss, %	2.33 ± 0.41	2.95 ± 0.30	3.05 ± 0.24	0.303
Cooking loss, %	34.87 ± 1.07	34.49 ± 0.82	34.67 ± 1.00	0.965
Marbling score	3.75 ± 0.22	3.25 ± 0.31	3.20 ± 0.15	0.189
45 min				
L*(lightness)	42.19 ± 0.47 ^A^	44.80 ± 0.50 ^B^	44.35 ± 0.37 ^B^	0.063
a* (redness)	8.27 ± 0.40	7.66 ± 0.39	7.39 ± 0.40	0.330
b* (yellowness)	6.77 ± 0.33	7.29 ± 0.19	7.18 ± 0.12	0.421
24 h				
L*(lightness)	51.41 ± 0.37	52.65 ± 0.30	52.06 ± 0.25	0.129
a* (redness)	11.88 ± 0.57 ^A^	12.54 ± 0.31 ^AB^	12.82 ± 0.26 ^B^	0.067
b* (yellowness)	9.53 ± 0.15 ^B^	9.96 ± 0.12 ^B^	8.74 ± 0.27 ^A^	0.052

Notes: Results are presented as the mean ± SEM (*n* = 6). Capital letters within the same line indicate 0.05 ≤ *p* < 0.10.

**Table 6 animals-12-02177-t006:** Effect of dietary yeast culture supplementation on blood metabolites of growing–finishing pigs.

Item	CON	GRA	YC	*p* Value
D 30				
Total protein, g/L	70.43 ± 3.45	68.33 ± 5.46	68.45 ± 3.63	0.927
Albumin, g/L	38.58 ± 1.81	35.90 ± 2.17	37.55 ± 0.73	0.539
GLU, mmol/L	4.23 ± 0.39	3.78 ± 0.35	5.13 ± 0.62	0.151
UN, mmol/L	6.95 ± 0.78	5.95 ± 0.69	6.70 ± 0.54	0.693
D 60				
Total protein, g/L	67.87 ± 3.02 ^a^	61.35 ± 2.81 ^ab^	55.00 ± 1.66 ^b^	0.048
Albumin, g/L	39.25 ± 2.47 ^a^	32.67 ± 2.23 ^ab^	28.28 ± 2.61 ^b^	0.041
GLU, mmol/L	4.25 ± 0.51 ^B^	3.14 ± 0.26 ^A^	3.04 ± 0.36 ^A^	0.085
UN, mmol/L	7.92 ± 0.73 ^a^	6.35 ± 0.73 ^ab^	4.84 ± 0.57 ^b^	0.038

Notes: GLU, glucose; UN, urea nitrogen. Results are presented as the mean ± SEM (*n* = 6). Small letters within the same line indicate *p* < 0.05 and capital letters indicate 0.05 ≤ *p* < 0.10.

**Table 7 animals-12-02177-t007:** Effect of dietary yeast culture supplementation on antioxidant capacity of finishing pigs.

Item	CON	GRA	YC	*p* Value
D 30				
CAT, U/mL	1.93 ± 0.29	1.68 ± 0.27	1.48 ± 0.28	0.557
T-AOC, U/mL	4.70 ± 0.29	5.05 ± 0.43	4.57 ± 0.37	0.643
T-SOD, U/mL	73.69 ± 1.52 ^a^	80.13 ± 1.86 ^b^	82.13 ± 0.49 ^b^	0.002
MDA, nmol/mL	3.08 ± 0.27	2.92 ± 0.19	3.02 ± 0.43	0.936
D 60				
CAT, U/mL	1.73 ± 0.16 ^a^	1.71 ± 0.43 ^a^	3.50 ± 0.36 ^b^	0.048
T-AOC, U/mL	4.68 ± 0.28	4.53 ± 0.23	4.04 ± 0.34	0.279
T-SOD, U/mL	82.79 ± 1.47	82.69 ± 1.99	79.13 ± 1.50	0.243
MDA, nmol/mL	2.50 ± 0.21 ^A^	2.74 ± 0.17 ^AB^	2.26 ± 0.19 ^B^	0.050

Notes: T-AOC, total antioxidant capacity; T-SOD, total superoxide dismutase; CAT, catalase; MDA, malonaldehyde. Results are presented as the mean ± SEM (*n* = 6). Small letters within the same line indicate *p* < 0.05 and capital letters indicate 0.05 ≤ *p* < 0.10.

**Table 8 animals-12-02177-t008:** Effect of dietary yeast culture supplementation on intestinal morphology of finishing pigs.

Items	CON	GRA	YC	*p* Value
Jejunum				
VH, μm	359.65 ± 17.10	346.78 ± 19.98	351.95 ± 24.60	0.908
CD, μm	108.94 ± 6.13	104.44 ± 10.54	91.32 ± 4.32	0.286
VH/CD	2.98 ± 0.24	3.42 ± 0.34	3.68 ± 0.05	0.104
Ileum				
VH, μm	370.16 ± 10.16	377.86 ± 13.79	330.29 ± 26.14	0.178
CD, μm	115.70 ± 7.71	117.15 ± 5.65	105.15 ± 8.84	0.489
VH/CD	3.26 ± 0.22	3.27 ± 0.25	3.24 ± 0.40	0.997

Notes: VH, villous height; CD, crypt depth. Results are presented as the mean ± SEM (*n* = 6).

## Data Availability

The data for 16S rRNA gene sequencing and RNA-seq are available in the National Center for Biotechnology Information under the accession numbers PRJNA814434. Data available on reasonable request from the corresponding author.

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
