# Peer review of "Effects of Yeast Culture Supplementation in Wheat–Rice-Based Diet on Growth Performance, Meat Quality, and Gut Microbiota of Growing–Finishing Pigs"

_animals, 2022, doi:10.3390/ani12172177_

Round 1

Reviewer 1 Report

This is a very interesting topic. With the rising prices of grain (feed), the economics of production increasing the profitability is extremely important. The analysis of another factor influencing the quality of pork is also very valuable. However, I have a few questions: 

LINE 56: There are also other co-authors

LINE 56-58: I Propose to add a sentence regarding "growth performance" - also an economically important indicator

LINE 90: Isn't that too few porkers? 15 per group may not be sufficient to calculate statistical differences.

LINE 90-91: “50.75 ± 1.319 kg body 90 weight (BW)]” shouldn't the experiment be started with a lower weight? For example, from the beginning of fattening? When is the most growth of intestinal villi?

LINE 100-103: The aim of the study is to evaluate the effect of YC addition to basic wheat and rice feed. Shouldn't the GRA group be the control group and the YC experimental group? If there are 3 groups left, it suggests redrafting the aim of the study, the more so because in the results, for example by comparing the cost of feed - the CON group is compared to the GRA group.

LINE 105: Table 1- no line below the heading in 4 last columns

LINE 123: dried? Rather, it is about cleansing rather than removing water from the organs?

LINE 171: Suggests including the GLM model in the manuscript

LINE 173: "tendency" - it is not visible in the tables, it can be marked with capital letters A, B, C ...

LINE 204: 45 min. L * (lightness) is also less than 0.1

LINE 211-212: why 31 and 61?

LINE 220- as above.

Table 3- 8 - Explanations under tables CON, GRA, YC are redundant, abbreviations already explained once in material and methods.

Figure 1- 6 - explanations under figures CON, GRA, YC are superfluous, abbreviations already explained once in material and methods.

LINE 291: Shouldn't family microbiom names be italic? There is italics in the discussion.

LINE 329: as above.

LINE 334-336: this is the conclusion

LINE 337: why 64 days?

LINE 349-352: unclear sentence. It appears to be about this research, but another article-8 is cited at the end

LINE 371-372: Better to give information about pork than lamb.

LINE 390: "The intestine plays an important role in animal health and growth." Each organ affects proper health, please specify

LINE 390-391: this conclusion - I proposes to modify this sentence

LINE 404: please specify what kind of birds they are

LINE 429: "dressing rate"? I propose to change

LINE 454: proposes to extend the conclusion with a few sentences.

Author Response

We thank the Editors and reviewers for their time and good suggestions.

The following is point-by-point response to the comments.

This is a very interesting topic. With the rising prices of grain (feed), the economics of production increasing the profitability is extremely important. The analysis of another factor influencing the quality of pork is also very valuable. However, I have a few questions: 

  1. LINE 56: There are also other co-authors

[Answer]: Thanks for your careful review, we have revised it.

  1. LINE 56-58: I Propose to add a sentence regarding "growth performance" - also an economically important indicator.

[Answer]: Thanks for your good suggestion. We have revised it.

  1. LINE 90: Isn't that too few porkers? 15 per group may not be sufficient to calculate statistical differences.

[Answer]: Thanks for your good suggestion. It is true that the more replication of each treatment, the more statistical differences will occur for the data analysis. 15 pigs per treatment are not many, so there are some data that can be seen to have changes, but the statistical analysis is not significant. If more replications can be used, the effect of yeast supplementation may be more obvious and significant.

Effect of yeast culture (Saccharomyces cerevisiae) supplementation on growth performance, blood metabolites, carcass traits, quality, and sensorial traits of meat from pigs under heat stress. Thirty pigs and ten pigs in each treatment were used in study, and found YC improved both growth and carcass weight (P<0.05).

  1. LINE 90-91: “50.75 ± 1.319 kg body 90 weight (BW)]” shouldn't the experiment be started with a lower weight? For example, from the beginning of fattening? When is the most growth of intestinal villi?

[Answer]: Thanks for good suggestion about the new experimental ideas. The longer the YC supplementation, the higher the animal feed cost. If the same effect can be achieved by supplemented with YC in the last month (from the beginning of fattening) or so, the farm income will be greatly improved. Meanwhile, the critical period of intestinal development is prenatal and early postnatal. In our study, 50kg pig were used and their gastrointestinal tract was already mature, which may also be the reason why the intestinal structure was not significantly different.

  1. LINE 100-103: The aim of the study is to evaluate the effect of YC addition to basic wheat and rice feed. Shouldn't the GRA group be the control group and the YC experimental group? If there are 3 groups left, it suggests redrafting the aim of the study, the more so because in the results, for example by comparing the cost of feed - the CON group is compared to the GRA group.

[Answer]: Thank for your good suggestion. This is good idea. The aim of the study is to compare the effect of different diet type on pig growth performance and meat quality. So, we think we should keep 3 groups, and revised the aim, and the results. The price of feed stuff such as corn and wheat fluctuate, therefore, we recommend when it is more reasonable to use wheat and rice. Please see the Line 182-184 .

  1. LINE 105: Table 1- no line below the heading in 4 last columns

[Answer]: Thanks for your carefulness. We have revised it.

  1. LINE 123: dried? Rather, it is about cleansing rather than removing water from the organs?

[Answer]: Sorry for our unclear description. We have revised it according to your good suggestion.

  1. LINE 171: Suggests including the GLM model in the manuscript

[Answer]: Thanks, we have revised it according to your good suggestion.

  1. LINE 173: "tendency" - it is not visible in the tables, it can be marked with capital letters A, B, C ...

[Answer]: Thank for your good suggestion. We have revised it in Tables.

  1. LINE 204: 45 min. L * (lightness) is also less than 0.1

[Answer]: Yes, it is very interesting that the L * (lightness) at 45 min in GRA and YC groups have the tendency to increasing compared with the CON group, but there is no significant difference or tendency at 24h.

  1. LINE 211-212: why 31 and 61?

[Answer]: Thanks for your careful review. We have revised it.

  1. LINE 220- as above.

[Answer]: Thanks for your careful review. We have revised it.

13.Table 3- 8 - Explanations under tables CON, GRA, YC are redundant, abbreviations already explained once in material and methods.

[Answer]: Thanks for your good suggestion, we have deleted this information in each table. 

  1. Figure 1- 6 - explanations under figures CON, GRA, YC are superfluous, abbreviations already explained once in material and methods.

[Answer]: Thanks, we have deleted this information in each figure. 

  1. LINE 291: Shouldn't family microbiom names be italic? There is italics in the discussion.

[Answer]: Thanks for your good suggestion, we have revised them.

  1. LINE 329: as above.

[Answer]: Thanks for your good suggestion, we have revised them.

  1. LINE 334-336: this is the conclusion

[Answer]: Sorry for our mistake, we have revised it.

  1. LINE 337: why 64 days?

[Answer]: Sorry for our unclear description, this is another research period. We have revised it.

  1. LINE 349-352: unclear sentence. It appears to be about this research, but another article-8 is cited at the end

[Answer]: Thanks for your good suggestion, we have revised it.

  1. LINE 371-372: Better to give information about pork than lamb.

[Answer]: Thanks for your good suggestion. Although there are still some literature studies on the effect of YC supplementation on pigs, the relationship between meat quality and antioxidant capacity is less discussed. This article is cited to illustrate that the improvement of meat quality after YC supplementation in diets is related to the antioxidant capacity.

  1. LINE 390: "The intestine plays an important role in animal health and growth." Each organ affects proper health, please specify

[Answer]: Sorry for our unclear description, we have revised it.

  1. LINE 390-391: this conclusion - I proposes to modify this sentence

[Answer]: Thanks for your good suggestion, we have modified this sentence. Hope hope it can express our meaning more clearly.

  1. LINE 404: please specify what kind of birds they are

[Answer]: Sorry for our unclear description, we have revised it.

  1. LINE 429: "dressing rate"? I propose to change

[Answer]: Thanks for your good suggestion, we have modified these words.

  1. LINE 454: proposes to extend the conclusion with a few sentences.

[Answer]: Thanks for your good suggestion, we have revised it, hope it will be better this time.

Reviewer 2 Report

Comments to the authors

Introduction 

·       L53-57: add appropriate references

·       L82: Saccharomyces cerevisiae (S. cerevisiae)

Materials and Methods

§  L91: How many female and male piglets per group? were there any sex differences in the results? please add this information

§  L96: BW

§  L120-123: add appropriate references

§  L182: BW in table 3

§  L234: increase the size of figure 1

Discussion

§  L336: Previous study has shown ….

§  L441-443: don’t use underlined words 

§  Add a paragraph about the economic impact of your results for swine industry and production cost 

Author Response

We thank the Editors and reviewers for their time and good suggestions.

The following is point-by-point response to the comments.

Review 2  Comments to the authors

Introduction 

  • L53-57: add appropriate references

[Answer]:

  • L82: Saccharomyces cerevisiae(S. cerevisiae)

[Answer]: Thanks for your good suggestion, we have added this information.

Materials and Methods

  • L91: How many female and male piglets per group? were there any sex differences in the results? please add this information.

[Answer]: This is a good suggestion. The female and male piglets per group is little difference. The number of females is 9, and the male is 6 in each group. In order to avoid gender differences, female pigs were selected for meat quality evaluation. We also added this information in the paper.

  • L96: BW

[Answer]: Thanks for your good suggestion, we have modified this word.

  • L120-123: add appropriate references

[Answer]: Thanks for your good suggestion, we have added it.

  • L182: BW in table 3

[Answer]: Thanks for your good suggestion, we have modified it.

  • L234: increase the size of figure 1

Discussion

  • L336: Previous study has shown ….

[Answer]: Thanks for your good suggestion, we have modified it.

  • L441-443: don’t use underlined words 

 [Answer]: Sorry for these errors, we have modified it. Thanks for your good suggestion.

  • Add a paragraph about the economic impact of your results for swine industry and production cost 

 [Answer]: This is a very good suggestion. The aim of this study is hope to decrease the feed cost. The price of corn and wheat fluctuates and changes at any time. If the prices of corn and wheat are equal, the cost-saving effect is not obvious. When the price of wheat is lower than 90% of that of corn, the application of wheat-rice-type diet will have a great effect on cost saving. In the first paragraph of discussion and conclusion, we added some information. Hope it will be better.

Round 2

Reviewer 1 Report

Please check the designations under the tables again. In the table 5 you only have capital letters, you don't have to write about "small letter" in the footer (legend) because they are not there. However, in table 7 you only have lowercase letters - do not write "capital letter" in the footer.

 In the chapter on statistical analysis, please provide a detailed model (GLM) of all factors included in the statistical analysis. It wasn't just about adding a abbreviation. 

Author Response

We thank the Editors and reviewers for their time and good suggestions.

The following is point-by-point response to the comments.

Review 2

Comments and Suggestions for Authors

  1. Please check the designations under the tables again. In the table 5 you only have capital letters, you don't have to write about "small letter" in the footer (legend) because they are not there.

[Answer]: Thanks for your carefulness and good suggestion. We have revised it.

  1. However, in table 7 you only have lowercase letters - do not write "capital letter" in the footer.

[Answer]: Thanks for your good suggestion. We have added this information in the footer.

  1. In the chapter on statistical analysis, please provide a detailed model (GLM) of all factors included in the statistical analysis. It wasn't just about adding a abbreviation. 

[Answer]: Thanks for your good suggestion. We have discussed this question. In our study, Three groups were designed to compare the effect of diet type and yeast culture supplementation. It will be better to use 4 groups (2 factor× 2 level design, CON, CON+YC, GRA, GRA+YC). Similar paper used the one-way analysis of variance (ANOVA) to test homogeneity of variances via Levene’s test and followed with Duncan’s test ( https://doi.org/10.1155/2020/1240152), while data were analyzed by by GLM, with pen as the experimental unit (doi:10.2527/jas.2007-0110, doi:10.1111/asj.12849). The following model was used: Yijk = μ + groupi + block within groupij + experimental treatmentk + eijk. where Yijk is the dependent variable, μ is the overall mean, group is the fixed effect of group , block within group is the fixed effect of block j nested within group i, experimental treatment is the fixed effect of experimental treatment , and eijk is the error.

In our research, treatment was the fixed effect. Covariates such as age / sex should be considered. But we choose female pigs to get blood sample and slaughter for meat quality analysis, the sex is same. After analyzing the data such as feed intake and daily gain, we found there is no sex effect.

For ages, we have two phase, 30, 60 days. Yij = μ+ experimental treatmenti + agej + experimental treatmenti × agej + eijk. After analyzing data, the age has significant effect on feed intake and daily gain because of their differences in growth stages of pigs, but there is no interaction between treatments and age.

Growth performance

Item

0-30d

SEM

31-60d

SEM

P Value

CON

GRA

YC

CON

GRA

YC

Diet

Phase

Diet*Phase

ADG/kg

1.08

1.07

1.08

0.02

1.06

1.08

1.05

0.04

0.918

0.447

0.702

ADFI/g

2960.69

2860.75

2926.50

71.36

3340.67

3389.89

3269.53

156.15

0.938

0.000

0.469

F/C

2.75

2.67

2.72

0.05

3.17

3.14

3.14

0.08

0.815

0.000

0.805

Because meat quality only has one phase, we keep the data.

Thanks for your good suggestion. Your suggestion provides good guidance and help for our future experiment design and experiment.

Reviewer 2 Report

None 

Author Response

We thank the Editors and reviewers for their time and good suggestions.